# Mutant p53 Depletion by Novel Inhibitors for HSP40/J-Domain Proteins Derived from the Natural Compound Plumbagin

**DOI:** 10.3390/cancers14174187

**Published:** 2022-08-29

**Authors:** Mohamed Alalem, Mrinalini Bhosale, Atul Ranjan, Satomi Yamamoto, Atsushi Kaida, Shigeto Nishikawa, Alejandro Parrales, Sana Farooki, Shrikant Anant, Subhash Padhye, Tomoo Iwakuma

**Affiliations:** 1Department of Pediatrics, Division of Hematology & Oncology, Children’s Mercy Research Institute, Kansas City, MO 64108, USA; 2Department of Cancer Biology, University of Kansas Medical Center, Kansas City, KS 66160, USA; 3Department of Chemistry, University of Pune, Pune 411007, India

**Keywords:** HSP40, DNAJA1, mutant p53, inhibitor, natural compound

## Abstract

**Simple Summary:**

The tumor suppressor p53 is frequently mutated in human cancer. Accumulation of missense mutant p53 (mutp53) in tumors is crucial for malignant progression, and cancers are often addicted to oncogenic mutp53. However, strategies to deplete mutp53 have not yet been established. Recent studies have shown that misfolded or conformational mutp53 is stabilized by DNAJA1, a member of HSP40, also known as J-domain proteins (JDPs). However, no selective DNAJA1 inhibitor is clinically available. Through a molecular docking study, we identified a potential DNAJA1 inhibitor, called PLTFBH, derived from the natural compound plumbagin, as a compound that bound to and reduced protein levels of DNAJA1 and several other HSP40/JDPs. PLTFBH reduced the levels of conformational mutp53 and inhibited cancer cell migration in a manner dependent on DNAJA1 and mutp53.

**Abstract:**

Accumulation of missense mutant p53 (mutp53) in cancers promotes malignant progression. DNAJA1, a member of HSP40 (also known as J-domain proteins: JDPs), is shown to prevent misfolded or conformational mutp53 from proteasomal degradation. Given frequent addiction of cancers to oncogenic mutp53, depleting mutp53 by DNAJA1 inhibition is a promising approach for cancer therapy. However, there is no clinically available inhibitor for DNAJA1. Our in silico molecular docking study with a natural compound-derived small molecule library identified a plumbagin derivative, PLIHZ (plumbagin–isoniazid analog), as a potential compound binding to the J domain of DNAJA1. PLIHZ efficiently reduced the levels of DNAJA1 and several conformational mutp53 with minimal impact on DNA contact mutp53 and wild-type p53 (wtp53). An analog, called PLTFBH, which showed a similar activity to PLIHZ in reducing DNAJA1 and mutp53 levels, inhibited migration of cancer cells specifically carrying conformational mutp53, but not DNA contact mutp53, p53 null, and wtp53, which was attenuated by depletion of DNAJA1 or mutp53. Moreover, PLTFBH reduced levels of multiple other HSP40/JDPs with tyrosine 7 (Y7) and/or tyrosine 8 (Y8) but failed to deplete DNAJA1 mutants with alanine substitution of these amino acids. Our study suggests PLTFBH as a potential inhibitor for multiple HSP40/JDPs.

## 1. Introduction

Discovery of novel therapeutic agents targeting cancer-specific events is a promising avenue for cancer treatment regimens [1]. The tumor suppressor *p53* (*p53*) is the most frequently mutated gene in human cancers [2,3]. Wild-type p53 (wtp53) functions as a transcription factor that inhibits tumor development by transactivating numerous downstream target genes involved in cell cycle arrest and cell death [4]. Most mutations in the *p53* gene are missense mutations that impair the function of wtp53 as a transcription factor [3]. Missense mutant p53 (mutp53) proteins frequently accumulate in cancer cells. Accumulated mutp53 not only inhibits wtp53′s tumor suppressive activity (dominant negative) but also shows oncogenic activities independent of wtp53 [5,6,7,8]. Overexpression of mutp53 in p53-null cells promotes cancer malignancy including metastasis and drug resistance [9]. These oncogenic activities of mutp53 are referred to as gain of function (GOF), which can be caused by mupt53′s ability to bind to other tumor-suppressive (e.g., p63, p73) and oncogenic (e.g., Ets2, SREBP2) proteins and alter the functions of these binding partners [9,10].

Mutations in p53 can roughly be classified into two types: DNA contact (class I) and misfolded or conformational (class II) types. DNA contact mutp53 has mutations within the amino acids that directly bind to DNA (e.g., R248W, R273H, R280K) without significant changes in the p53 structure, while the misfolded or conformational mutp53 has mutations that lead to robust alterations of the p53 protein structure (e.g., R156P, R175H, Y220C) [11]. Both types of missense mutations in p53 result in loss of the tumor suppressive activity [3]. Importantly, genetic or pharmacologic depletion of mutp53 results in reduced malignant properties of cancer cells, suggesting that cancer cells are addicted to mutp53 [12,13]. Thus, understanding the mechanism(s) of mutp53 stabilization or degradation could help develop novel strategies for depleting oncogenic mutp53 and consequently inhibiting cancer progression.

The mechanisms underlying mutp53 stability in cancer cells are still not fully understood. While wtp53 is mainly ubiquitinated by MDM2, mutp53 is ubiquitinated and degraded by multiple ubiquitin ligases including MDM2, COP1, Pirh2, and CHIP [14]. Our group recently reported that DNAJA1, a member of heat shock protein 40 (HSP40) family, also known as J-domain proteins (JDPs), can stabilize mainly misfolded or conformational type of mutp53 [15]. HSP40/JDPs are molecular chaperones that contain a common J domain comprising 70 amino acid residues. HSP40/JDPs regulate proteostasis and various cellular activities including protein translation, folding/unfolding/refolding, and stabilization/degradation [16,17,18]. Knockdown of DNAJA1 results in CHIP ubiquitin ligase-mediated nuclear export, ubiquitination, and proteasomal degradation of misfolded or conformational mutp53 while having minimal impact on the levels of wtp53 and DNA contact mutp53 [15]. Based on these observations, we hypothesize that inhibitors of DNAJA1 would specifically reduce the levels of misfolded or conformational mutp53, inhibiting malignant properties of cancer cells. Such inhibitors may cause minimal side effects, since normal cells do not commonly carry mutp53. Recently, Moses et al. [19] identified a chalcone compound, C86, as a small molecule that induced protein degradation of full-length and variant androgen receptors through binding to several HSP40/JDPs including DNAJB1. However, it remains unclear if C86 could functionally inhibit DNAJA1 and induce mutp53 degradation. Moreover, no specific inhibitors for DNAJA1 or HSP40/JDPs are commercially or clinically available. To test our hypothesis and identify potential DNAJA1 inhibitors, we have performed an in silico molecular docking study for the J domain of DNAJA1 using a natural compound-derived small molecule library. We identified that plumbagin derivatives bound to DNAJA1 in cells and reduced the levels of DNAJA1 and other HSP40/JDPs, leading to decreased protein levels of multiple conformational mutp53 and reduced cancer cell migration in a manner dependent on DNAJA1 and mutp53.

## 2. Materials and Methods

### 2.1. Cell Lines

Cell lines with different status of p53 were cultured and maintained in Dulbecco’s modified Eagle’s medium (DMEM, Fisher Scientific, Pittsburgh, PA, USA) with 10% fetal bovine serum (FBS, Neuromics, Edina, MN, USA), 1% penicillin–streptomycin (Fisher Scientific, Pittsburgh, PA, USA) and 0.1% normocin (Invivogen, San Diego, CA, USA). All cell lines used are routinely maintained in the laboratory. The cell lines with conformational mutp53 include human pharyngeal squamous cell carcinoma HN31 (p53^C176F^), tongue squamous cell carcinoma CAL33 (p53^R175H^), and osteosarcoma KHOS/NP (p53^R156P^). Cell lines with DNA contact mutp53 include breast adenocarcinoma MDA-MB-231 (p53^R280K^), pharyngeal squamous cell carcinoma FaDu (p53^R248L^), colorectal adenocarcinoma HT29 (p53^R273H^), and pancreatic cancer Panc-1 (p53^R273H^). Cell lines with wtp53 include osteosarcoma SJSA1 (p53^wt^) and U2OS (p53^wt^), as well as pharyngeal cancer HN30 (p53^wt^). p53-null cell lines include non-small cell lung cancer H1299 (p53^null^), squamous cell carcinoma of the oral tongue OSC-19 (p53^null^), and tongue squamous cell carcinoma SAS (p53^null^). Wtp53-bearing non-tumor cell lines include normal human esophageal squamous cell Het-1a (p53^wt^) and normal human fibroblasts, BJ and WI38 (p53^wt^). U2OS, SJSA1, KHOS/NP, HT29, Panc-1, H1299, WI38, BJ, and Het1A were obtained from ATCC. SAS was obtained from A.K. at the Tokyo Medical & Dental University, while CAL33, HN31, FaDu, OSC-19, and HN31 were kindly obtained from Dr. Sufi Thomas at the University of Kansas Medical Center [20]. The authenticity of all these cell lines was verified by autosomal short tandem repeat (STR) genotyping analysis services provided by the University of Arizona Genetics Core. None of these cells were reported as being misidentified by National Center for Biotechnology Information (NCBI) and International Cell Line Authentication Committee (ICLAC).

### 2.2. Molecular Docking Studies

The protein structure [2LO1] is obtained from RCSB Protein Data Bank (PDB), and the 3D structure of PLIHZ was obtained using the CORINA 3D structure generator (https://www.macinchem.org/blog/files/d8e06d427263c87a311969a753f9bf8c-1020.php) (accessed on 14 September 2015). The compounds were docked into the cavity of the J domain of DNAJA1 using the AutoDock Vina software [21]. Docking results were visualized using the PyMOL viewer (https://pymol.org/2/) (accessed on 14 September 2015).

### 2.3. Chemicals and Compounds

Protease/phosphatase inhibitors were purchased from Pierce^TM^/ThermoFisher Scientific (Waltham, MA, USA). The 3-(4,5-dimethylthiazol-2-yl)-2,5-diphenyltetrazolium bromide (MTT) reagent and mounting media (ProLong™ Gold Antifade Mountant with 4′,6-diamidino-2-phenylindole DAPI) were purchased from Invitrogen/Thermo Scientific (Waltham, MA, USA). Triton-X 100 and bovine serum albumin (BSA) were purchased from Sigma-Aldrich, Inc. (St. Louis, MO, USA). Color Prestained Protein Standard, Broad Range (10–250 kDa), was purchased from New England Biolabs (P7719S, Ipswich, MA, USA). Trypan blue (Tolidine-disazo-bis-8-amino-1-naphthol-3,6-disulfonic acid tetrasodium salt) was purchased from Sigma-Aldrich. Phalloidin actin stain, rhodamine conjugate, was purchased from Biotium (00027, Fremont, CA, USA). Puromycin was purchased from Invivogen (San Diego, CA, USA). G418 disulfate salt was purchased from Sigma-Aldrich, Inc. JetPrime transfection reagent was purchased from Polyplus-transfection (New York, NY, USA). Dimethyl sulfoxide (DMSO) was purchased from Sigma-Aldrich, Inc. PLIHZ compound and its analogs were synthesized and chemically characterized by our collaborator Dr. Padhye, a visiting professor at University of Kansas Medical Center and a faculty member at the University of Pune, Maharashtra, India [22]. Chemical code of each compound is PLIHZ (plumbagin-isonicotiyl hydrazide), PLFBH (plumbagin-flurobenzoic hydrazide), PLTFBH (plumbagin-triflurobenzoic hydrazide), PLFUH (plumbagin-2-furoic hydrazide), and PLOCT (plumbagin-octanoic hydrazide).

### 2.4. Plasmids

A lentiviral vector encoding human *DNAJA1* (HsCD00437433), pLX304-hDNAJA1, was purchased from (DNASU Plasmid Repository in Arizona State University, Tempe, AZ, USA). Full-length (FL) *DNAJA1* cDNA (GenBank accession number EU176556) was amplified by PCR and inserted into BamHI and NotI restriction enzyme sites of the pCDH-CMV-MCS-EF1-puro vector purchased from System Biosciences (Palo Alto, CA, USA). In addition to the constructs containing FL-DNAJA1, constructs with DNAJA1-Y7A-, and Y8A-mutants were generated using the QuikChange II XL Site-Directed Mutagenesis Kit (Agilent, Santa Clara, CA, USA). All coding regions were verified by DNA sequencing (ACGT, Inc., Wheeling, IL, USA). A lentiviral vector encoding a *DNAJA1* shRNA (TRCN0000275847, Clone ID: NM_001539.2-1078s21c) was purchased from Sigma-Aldrich, Inc and a lentiviral vector encoding *p53* shRNA (shp53 pLKO.1 puro, #19119) was purchased from Addgene (Watertown, MA, USA), while pGIPz control encoding-vectors were purchased from Open Biosystems, Inc. (Huntsville, AL, USA). *DNAJA1* sgRNA CRISPR lentivector (#2, target sequence: GAGTGCTGTCCCAATTGCCG) from ABM, Inc. (Richmond, BC, Canada) and *p53* sgRNA CRISPR lentivector (pXPR003-sgTP53-4, #118022, target sequence: CCCCGGACGATATTGAACAA) from Addgene. *GFP-Cas9*-encoding adenoviral vector was purchased from VECTOR BIOLABS (1901, Malvern, PA, USA).

### 2.5. Generation of DNAJA1 and p53 Knockout or Knockdown Cell Lines

*DNAJA1*- or *p53*-knockout HN31 cells were generated according to the protocol described previously [20]. HN31 cells were infected with scramble control, *DNAJA1* sgRNA-, or *p53* sgRNA-encoding lentivectors, followed by infection with the *GFP-Cas9*-encoding adenoviral vector. Cells were selected by G418 or puromycin, followed by single colonization. Deletions of *DNAJA1* or *p53* genes were confirmed by Western blotting and then verified by DNA sequencing. Knockdown of DNAJA1 and p53 was performed using their specific short hairpin RNAs (shRNAs) as previously described [15].

### 2.6. Antibodies

The following antibodies were used: mouse monoclonal anti-p53 (sc-126, DO-1, Santa Cruz Biotechnology, Inc., Dallas, TX, USA), rabbit monoclonal anti-p53 (#2527, 7F5, Cell Signaling Technology, Danvers, MA, USA), anti-DNAJA1 (KA2A5.6, Invitrogen/Thermo Fisher Scientific, Carlsbad, CA, USA), anti-DNAJA1 (11713-1-AP, Proteintech, Rosemont, IL, USA), anti-DNAJA2 (12236-1-AP, Proteintech), anti-DNAJA3 (HPA040875, Santa Cruz Biotechnology), anti-DNAJA4 (LL2, sc-100714, Santa Cruz Biotechnology, Inc.), anti-DNAJB1 (13174-1-AP, Proteintech), anti-DNAJB2 (10838-1-AP, Proteintech), anti-DNAJB6 (11707-1-AP, Proteintech), anti-DNAJB12 (16780-1-AP, Proteintech), anti-DNAJC2 (11971-1-AP, Proteintech), anti-DNAJC3 (A-7, sc-393559, Santa Cruz Biotechnology, Inc.), anti-DNAJC6 (21941-1-AP, Proteintech), anti-DNAJC7 (11090-1-AP, Proteintech), anti-DNAJC8 (A301-839A, Bethyl Laboratories Inc./Fortis Life Sciences, Waltham, MA, USA), anti-DNAJC10 (3101-1-AP, Proteintech), anti-DNAJC15 (16063-1-AP, Proteintech), anti-DNAJC20 HSCB (15132-1-AP, Proteintech), anti-Rac1 (66122-1-Ig, Proteintech), anti-Cdc42 (ab64533, Abcam plc, Waltham, MA, USA), anti-GAPDH (H-12, Santa Cruz Biotechnology, Inc.), anti-vinculin (V284, Fitzgerald Industries International, Acton, MA, USA), IRDye 680RD donkey anti-rabbit IgG (926-68073, LI-COR Biosciences, Lincoln, NE, USA), and IRDye 800CW donkey anti-mouse IgG (926-32212, LI-COR Biosciences). Donkey anti-rabbit IgG (H+L) Highly Cross-Adsorbed Secondary Antibody, Alexa Fluor™ 568 (A10042) and donkey anti-mouse IgG (H+L) Highly Cross-Adsorbed Secondary Antibody, Alexa Fluor™ 488 (A21202) were purchased from Invitrogen/Thermo Scientific.

### 2.7. Western Blotting

Cells were lysed using CelLytic M buffer (Sigma-Aldrich, Inc.) containing protease inhibitors (Thermo Fisher Scientific). Equal amounts of cell lysate were separated on a tris-glycine gel (Bio-Rad Laboratories, Hercules, CA, USA), transferred to polyvinylidene fluoride (PVDF) membranes (Amersham™ Hybond^®^ P 0.45 μm, Cytiva, Global Life Sciences Solutions LLC, Marlborough, MA, USA), and blotted with designated primary antibodies for each protein of interest, followed by appropriate fluorescence-tagged secondary antibodies. The blots were analyzed with the Li-Cor Odyssey infrared imaging system (Li-Cor) or the Sapphire Biomolecular Imager (Azure Biosystems Inc., Dublin, CA, USA). Full images of the Western blots with molecular weight markers, as well as the densitometry intensity ratio of each band, are presented in Appendix A–full images of Western blots.

### 2.8. Immunofluorescence

Cells were seeded onto Lab Tek II Chamber Slides coated with poly-D-lysine mimic (154941, Thermo Fisher Scientific). After treatment with DMSO or compounds, cells were washed with PBS and fixed with 4% paraformaldehyde for 20 min. Then, cells were permeabilized with PBS containing 0.3% Triton X-100 (PBS-T) for 20 min. After blocking in 1% BSA in PBS-T for 1 h, cells were incubated with the designated primary antibodies, followed by fluorescence-tagged corresponding secondary antibodies. Samples were mounted using the ProLong Gold Antifade Reagent with DAPI (ThermoFisher). Images were analyzed using BZ-X800 Keyence All-in-One fluorescent microscope (KEYENCE CORPORATION, Itasca, IL, USA).

### 2.9. F-Actin Staining

Cells were seeded onto 24-well plates containing poly-L-lysine-coated 12 mm coverslips purchased from Corning (354085, Glendale, AZ, USA). Cells were treated with DMSO or PLTFBH for 16 or 24 h, fixed in 4% paraformaldehyde for 20 min, permeabilized with PBS-T for 20 min, and blocked in 1% BSA in PBS-T for 1 h. Then, the cells were incubated with phalloidin stain for 2 h, with gentle shaking, at room temperature. Quantification of filopodia formation at the circumference of cells was performed as previously described [20].

### 2.10. Transwell Migration Assay

Cells were pre-incubated with DMSO or PLTFBH for 12 h. The cells were then split, and viable cells were counted by trypan-blue staining. Viable cells (1–4 × 10^4^) suspended in DMEM with 0.5% FBS containing either DMSO or PLTFBH were placed in the upper permeable cell culture inserts (24-well plate, 8.0 µm pore size, CELLTREAT Scientific Products, Pepperell, MA, USA). The bottom chamber containing DMEM with 10% FBS was used as a chemoattractant. After 12 h, migrating cells on the bottom of the membrane were fixed, permeabilized, and stained with Diff-Quik Set (Dade Behring, Deerfield, IL, USA). Migrating cells were viewed using the EVOS M5000 microscope (Thermo Fisher Scientific). The numbers of migrating cells in the entire field were counted, and the percentage of the migrating cells was calculated compared to the control group.

### 2.11. Cellular Thermal Shift Assay (CETSA)

Cells treated with 80 µM of DMSO or compounds for 4 h were harvested in PBS and aliquoted into thin-walled PCR tubes (Thermo Fisher Scientific), followed by incubation at different temperatures, from 40 °C to 58 °C, for 3 min. After heating, cells were incubated at room temperature for 3 min and then subjected to 3 cycles of freeze–thaw to extract proteins. Following centrifugation to precipitate insoluble aggregated proteins, the supernatants, which contained compound-bound target proteins resistant to heat-induced denaturation, were used for Western blotting for different HSP40/JDPs.

### 2.12. Rac1/Cdc42 Pull-Down Activation Assay

The Rac1/Cdc42 pull-down activation assay kit was purchased from (PAK02, Cytoskeleton, Inc., Denver, CO, USA). Cells treated with DMSO or PLTFBH at ~1/2 IC_50_ (24 h) for 16 h were lysed in the manufacturer’s lysis buffer containing protease inhibitors. Cell lysates were centrifuged at 4 °C for 10 min, and the supernatant was incubated with the PAK-PBD beads at 4 °C for 2 h with rotation. After pelleting and washing the beads three times in ice-cold PBS, the beads were resuspended in SDS lysis buffer for Western blotting to detect active Rac1 and Cdc42.

### 2.13. Statistical Analysis

The differences between samples were analyzed by two-tailed Student’s t-test or two-way Analysis of Variance (ANOVA) using the GraphPad Prism 9 (GraphPad Software, Inc., San Diego, CA, USA). Data from at least three biological replicates were expressed as mean ± SEM, and the differences were considered statistically significant by *p* < 0.05.

## 3. Results

### 3.1. Knockdown of DNAJA1 Specifically Reduces Protein Levels of Conformational mutp53, but Not DNA Contact mutp53 and wtp53

Recent studies by our group and others demonstrate that mutp53, specifically misfolded or conformational mutp53, is stabilized by DNAJA1, a member of HSP40/JDPs [15,20,23,24]. We therefore validated this finding by knocking down DNAJA1 and examining protein levels of DNAJA1 and p53 in multiple cancer cell lines with different p53 status. These include HN31 (p53^C176F^, conformational), CAL33 (p53^R175H^, conformational), KHOS/NP (p53^R156P^, conformational), MDA-MB-231 (p53^R280K^, DNA contact), HT29 (p53^R273H^, DNA contact), FaDu (p53^R248L^, DNA contact), U2OS (wtp53), SJSA1 (wtp53), and HN30 (wtp53). Western blotting showed that knockdown of DNAJA1 specifically reduced protein levels of conformational mutp53 with minimal effects on the levels of DNA contact mutp53 and wtp53 (Figure 1A). These results were consistent with the results of immunofluorescence studies (Figure 1B and Appendix A). These results strongly suggest that DNAJA1 could be a therapeutic target for cancer therapy by specifically depleting conformational mutp53.

### 3.2. Identification of a Compound That Binds DNAJA1 to Specifically Reduce Conformational mutp53

Currently, no specific DNAJA1 inhibitors are commercially and clinically available. To identify a potential inhibitor for DNAJA1, we conducted an in silico docking study using a natural compound library against the J domain of DNAJA1 (PDB: 2LO1) and identified a plumbagin-derived PLIHZ as a compound that could bind to DNAJA1. The study also identified tyrosine 7 (Y7) as a critical amino acid for the PLIHZ-DNAJA1 interaction with binding energy of −6.6 kcal/mol at 2.6 Å distance (Figure 2A). PLIHZ was previously identified as a compound derived from a phytochemical plumbagin [22,25]. Although plumbagin regulates Akt signaling and p53 activity to show anti-tumor effects in multiple types of cancer [22,26,27,28], the biological effects of its derivative PLIHZ and the underlying mechanisms remain unclear.

To validate the DNAJA1-PLIHZ binding within cells, we performed a cellular thermal shift assay (CETSA), in which a compound-bound target protein becomes resistant to heat-induced protein denaturation and aggregation, causing the protein-compound complex to be retained in the soluble fraction after centrifugation of cell lysates [29,30,31]. In both CAL33 and HN31 cells, DNAJA1 protein levels were significantly increased between 46 °C and 55 °C in the supernatants of cells treated with PLIHZ as compared to the vehicle control (Figure 2B and Appendix A). These results suggest increased thermal stability of DNAJA1 protein by PLIHZ, thereby demonstrating physical binding between PLIHZ and DNAJA1 in cells.

Next, we determined the 24h-IC_50_ values of PLIHZ in multiple cancer and non-tumor cell lines with different p53 status. Cells harboring conformational mutp53 exhibited lower 24h-IC_50_ values for PLIHZ, compared to cells with DNA contact mutp53, p53 null, or wtp53, regardless of cancer cells or non-tumor cells (Appendix A). Hereafter, we used approximately half (1/2) of the IC_50_ values for experiments unless otherwise noted. We then treated cancer cells with different p53 status, including HN31, MDA-MB-231, U2OS, and H1299, with ~1/2 24h-IC_50_ of PLIHZ. We found that PLIHZ reduced protein levels of conformational mutp53 in HN31 cells with little effect on the p53 levels in MDA-MB-231, U2OS, and H1299 cells by Western blotting (Figure 2C). Intriguingly, PLIHZ reduced the DNAJA1 levels in all cell lines examined. Essentially the same results were observed using immunofluorescence studies (Figure 2D and Appendix A). These findings suggest that PLIHZ likely binds to DNAJA1 in all cells to reduce the protein level of DNAJA1; however, the reduction in DNAJA1 protein only alters the level of conformational mutp53, consistent with the results in Figure 1.

### 3.3. PLTFBH, An Analog of PLIHZ, Specifically Reduces Conformational mutp53 Levels Similar to PLIHZ

To improve the biological and chemical properties of PLIHZ, we synthesized four PLIHZ analogs, called PLFBH, PLTFBH, PLFUH, and PLOCT (Figure 3A). We compared the efficacy of PLIHZ and these analogs on reducing the protein level of conformational mutp53 by Western blotting. All analogs, except PLOCT, exhibited comparable potency in reducing the protein levels of DNAJA1 and conformational mutp53 in HN31 cells, as well as CAL33 and KHOS/NP cells (Figure 3B and Appendix A). Importantly, none of these analogs reduced the protein levels of wtp53 and DNA contact mutp53 (R248L, R273H) in SJSA1, HN30, FaDu, and HT29 cells (Appendix A). When we determined 72h-IC_50_ for these PLIHZ analogs, all analogs, except PLOCT, showed similar cytotoxic activities in cancer cell lines harboring conformational mutp53 including HN31, KHOS/NP, and CAL33 (Appendix A). Although PLIHZ, PLFBH, PLTFBH, and PLFUH showed comparable conformational mutp53-depleting and cytotoxic activities to cancer cells, PLTFBH consistently reduced the levels of multiple conformational mutp53 better than other analogs with minimal impact on wtp53 and DNA contact mutp53 (Figure 3B and Appendix A). Hence, hereafter, we focused on the PLTFBH compound.

Next, we determined 24h-IC_50_ values of PLTFBH in multiple cancer and non-tumor cell lines with different p53 status. Similar to PLIHZ, cells expressing conformational mutp53 showed relatively lower 24h-IC_50_ values for PLTFBH, compared to those expressing DNA contact mutp53, p53 null, or wtp53 (Appendix A). We then examined the effects of PLTFBH at 1/2 of 24h-IC_50_ on the levels of DNAJA1 and p53 in cancer cell lines with different p53 status. Similar to PLIHZ, PLTFBH reduced the levels of only conformational mutp53, although it decreased DNAJA1 levels in all cell lines examined by Western blotting (Figure 3C) and immunofluorescence (Figure 3D). We also confirmed the intracellular interaction of PLTFBH with DNAJA1 in CAL33 and HN31 cells by CETSA (Figure 3E and Appendix A).

To further explore dependency of the cytotoxic effects of PLTFBH on DNAJA1 and mutp53, we genetically deleted DNAJA1 or mutp53 in HN31 cells by the CRISPR-Cas9 strategy (Figure 3F). These HN31 sub-cell lines were treated with different concentrations of PLTFBH for 72 h, followed by MTT assays (Figure 3G). Although deletion of DNAJA1 or mutp53 made HN31 cells slightly more resistant to PLTFBH, the differences in the 72h-IC_50_ values were not statistically significant. These data suggest that the cytotoxic effect of PLTFBH is not entirely dependent on DNAJA1 and mutp53, and PLTFBH may have other biological targets in addition to DNAJA1.

### 3.4. PLTFBH Inhibits Migratory Potential of Cancer Cells in a Manner Dependent on DNAJA1 and Conformational mutp53

One of the major mutp53 GOF activities is to enhance cancer cell migration and metastasis [9,10,32,33]. Moreover, depletion of DNAJA1 results in reduced filopodia formation and migratory potential of cancer cells expressing conformational mutp53 [15,20]. To examine the effect of PLTFBH on DNAJA1- and conformational mutp53-dependent migration, we measured the migratory potential of cells with different p53 status, following PLTFBH treatment. As expected, PLTFBH significantly inhibited the migration of cancer cells harboring conformational mutp53 (HN31, KHOS/NP). In contrast, the migratory potential of cells harboring DNA contact mutp53 (MDA-MB-231, Panc-1), wtp53 (U2OS, SJSA1, Het-1a), and p53 null (H1299, SAS) was not altered by PLTFBH (Figure 4A). We also examined the effects of PLTFBH on filopodia formation of cancer cells with different p53 status. Consistently, PLTFBH inhibited filopodia formation in HN31 and CAL33 cells harboring conformational mutp53, whereas it had minimal impact on filopodia formation in cancer cells with DNA contact mutp53, wtp53, and p53 null (Figure 4B and Appendix A). Moreover, knockdown of either DNAJA1 or conformational mutp53 abrogated the PLTFBH-mediated inhibition of migration of HN31 and KHOS/NP cells (Figure 4C and Appendix A).

The signaling involved in filopodia formation and migration is regulated by the activity of Cdc42 and Rac1 [20,34,35]. Moreover, knockdown of DNAJA1 inhibits the activity of Cdc42 and Rac1 in HNSCC cells [20]. Hence, we examined the effects of PLTFBH on the Cdc42/Rac1 activities in HN31 and MDA-MB-231 cells. In agreement with the results of inhibited filopodia formation and migration by PLTFBH, PLTFBH reduced the active forms of Cdc42 and Rac1 in HN31 cells with a conformational mutp53, whereas it had minimal effects on the Cdc42 and Rac1 activities in MDA-MB-231 cells carrying a DNA contact mutp53 (Figure 4D and Appendix A). Together, these results strongly suggest that PLTFBH inhibits migratory potential of cancer cells predominantly in a manner dependent on DNAJA1 and mutp53, demonstrating specificity of PLTFBH for these targets.

### 3.5. PLTFBH Selectively Decreases Protein Levels of Certain Members of HSP40/JDPs

Although we show specificity of PLTFBH for inhibiting conformational mutp53-dependent cancer cell migration, PLTFBH still inhibited viable cell proliferation of HN31 cancer cells lacking DNAJA1 and mutp53 (Figure 3G), suggesting the possibility that PLTFBH had other targets than DNAJA1. All HSP40/JDPs members have well-conserved J domains and are classified to three classes of A, B, and C, based on their structures including a Gly-Phe rich region, C-terminal β-barrel domains, and a dimerization domain [20,36]. Therefore, we hypothesized that PLTFBH binds to and reduces protein levels of other HSP40/JDPs. To test this hypothesis, we examined the effect of PLTFBH on the protein levels of multiple HSP40/JDPs members, including DNAJA1, DNAJA2, DNAJA3, DNAJA4, DNAJB1, DNAJB2, DNAJB6, DNAJB12, DNAJC2, DNAJC3, DNAJC6, DNAJC7, DNAJC8, DNAJC10, DNAJC15, and DNAJC20, in HN31 cells. Besides DNAJA1, PLTFBH decreased the level of several other HSP40/JDPs to variable extents (Figure 5A). These results were further confirmed by immunofluorescence studies (Appendix A). We arbitrary classified their responses into three groups of good (DNAJA1, DNAJA2, DNAJA3, DNAJB1, DNAJB12, and DNAJC3), moderate (DNAJA4, DNAJB2, DNAJB6, DNAJC2, DNAJC7, DNAJC10, and DNAJC20), and little or no (DNAJC6, DNAJC8, and DNAJC15) (Figure 5B).

We also excluded the possibility that DNAJA1 depletion could lead to reduced protein levels of some of HSP40/JDPs. To address this concern, we examined the effects of *DNAJA1* knockout on the protein levels of several PLTFBH-responding HSP40/JDPs (DNAJB1, DNAJB12, DNAJC3, DNAJC7), as well as a non-responding DNAJC6, using control and *DNAJA1*-knockout HN31 cells. There was no change in the protein levels of these HSP40/JDPs by *DNAJA1* knockout, confirming that reduced protein levels of some HSP40/JDPs were not due to DNAJA1 depletion (Appendix A).

Our docking analyses identified Y7 in the J domain of DNAJA1 as a putative amino acid critical for the PLIHZ-DNAJA1 binding. Intriguingly, HSP40/JDPs lacking both Y7 and its neighboring amino acid Y8 (DNAJC6, DNAJC8, DNAJC15) failed to respond to PLTFBH, while PLTFBH reduced the protein levels of HSP40/JDPs with either or both of these amino acids to variable extents (Figure 5C). We next performed CETSA to determine whether PLTFBH could bind with DNAJA4 (moderate responder with both Y7 and Y8) and/or DNAJC6 (little or no responder lacking both Y7 and Y8) with PLTFBH using CAL33 cells. Consistent with the results above, PLTFBH successfully bound to DNAJA4; however, it failed to bind to DNAJC6 (Figure 5D). These results may suggest that the protein structure near residues Y7 and Y8 is crucial for PLTFBH’s binding to the J domain of HSP40/JDPs, resulting in depletion of the target proteins.

### 3.6. Mutations at Y7 and Y8 Residues in DNAJA1 Abrogate the Ability of PLTFBH to Deplete DNAJA1 and Conformational mutp53

To further delineate the significance of Y7 and Y8 in DNAJA1 for the action of PLTFBH, we re-expressed wild-type DNAJA1 (wt), a mutant DNAJA1 with Y7 substituted to alanine (Y7A), or a mutant DNAJA1 with Y8 substituted to alanine (Y8A) in *DNAJA1*-knockout HN31 cells, followed by Western blotting (Figure 6A) and immunofluorescence (Figure 6B). In the absence of DNAJA1, PLTFBH showed little effect on the protein level of endogenous conformational mutp53 (p53^C176F^). Re-introduction of wt or mutant DNAJA1 (Y7A, Y8A) resulted in rescue of mutp53 protein levels. PLTFBH treatment successfully reduced exogenous wild-type DNAJA1 (wt) and endogenous mutp53 protein levels. However, it failed to reduce exogenous Y7A and Y8A mutant DNAJA1. Accordingly, mutp53 levels in these mutant DNAJA1-expressing cells were unchanged (Figure 6A,B). We additionally confirmed reduction in endogenous DNAJB6 proteins by the PLTFBH treatment in these HN31 sub-cell lines (Appendix A).

We furthermore assessed the effects of mutant DNAJA1 (Y7A, Y8A), which rescued the endogenous mutp53 protein levels but did not respond to PLTFBH, on filopodia formation using these HN31 sub-cell lines (Figure 6C). Exogenous expression of wt or mutant DNAJA1 restored filopodia formation, consistent with the restored endogenous mutp53 levels. As expected, PLTFBH inhibited the filopodia formation induced by wt DNAJA1; however, it had minimal impact on the filopodia formation induced by Y7A and Y8A DNAJA1 mutants. Taken together, these results furthermore confirmed critical roles of Y7 and Y8 of DNAJA1 in PLTFBH-mediated depletion of DNAJA1 and conformational mutp53, leading to suppression of filopodia formation and migration.

## 4. Discussion

Our in silico docking-based study has identified a plumbagin-derivative PLIHZ and its analog PLTFBH as compounds that bind to the J domain of DNAJA1 and induce depletion of multiple conformational mutp53. PLIHZ is a synthetic naphthoquinone derived from the combination of the natural phytochemical agent, plumbagin, and the anti-tuberculosis agent, isonicotinic hydrazid (INH) [25]. Plumbagin is isolated from the roots of the medicinal plant *Plumbago zeylanica* and has been suggested as an anti-cancer, anti-inflammatory, and cytotoxic agent [26,37]. Plumbagin shows its anti-tumor effects through induction of cell cycle arrest, apoptosis, and autophagy, as well as inhibition of EMT by inhibition of Akt signaling, activation of wtp53, inhibition of NF-kB activity, and other unknown mechanisms, in multiple types of cancer [22,26,27,28]. However, the effects of the plumbagin analog PLIHZ on cellular signaling and cancer progression have not been tested [22,25].

We confirmed the intracellular binding of PLIHZ and its derivative PLTFBH to DNAJA1 by CETSA and their effects on reducing protein levels of conformational mutp53, but not wtp53 and DNA contact mutp53. Intriguingly, both PLIHZ and PLTFBH reduced the protein levels of DNAJA1 as well, although the underlying mechanism remains to be determined. Unfortunately, these compounds showed non-specific inhibition of viable cell proliferation, based on their cytotoxic effects in *DNAJA1* and/or *mutp53* knockout cells; however, PLTFBH showed specific inhibition of filopodia formation and migration of cancer cells expressing conformational mutp53 with minimal impact on cells with wtp53, p53 null, and DNA contact mutp53. Moreover, PLTFBH had minimal impact on the migration of cells lacking DNAJA1 and mutp53, demonstrating the on-target effects. Thus, to the best of our knowledge, this is the first study showing the efficient depletion of mutp53 and subsequent inhibition of cancer cell migration through inhibition or depletion of DNAJA1 by a natural product-derived compound.

Previously, Moses et al. [19] reported a potential HSP40/JDP inhibitor, chalcone C86, that induced degradation of androgen receptor (AR) and its variant ARv, similar to HSP70 inhibitors (JG98, JG231), by binding to the J domain of multiple HSP40/JDP members. Intriguingly, C86 does not alter the levels of HSP40/JDPs, unlike the case of PLIHZ or PLTFBH. Since their study does not use *HSP40/JDPs* knockout/knockdown cells, it remains unclear if the observed inhibition of HSP40/JDPs by C86 is the direct cause of depletion of AR and ARv and which HSP40/JDPs play roles in this activity. Additionally, side-by-side comparison studies of C86 and PLTFBH for their efficacy to reduce mutp53 levels would be important as a future study.

One major caveat associated with PLTFBH treatment is the induction of some cytotoxicity in cancer cells lacking DNAJA1 or mutp53. This observation may suggest that other HSP40/JDP proteins whose activities or levels are reduced by PLTFBH could regulate cancer cell proliferation or survival. Hence, DNAJA1- and mutp53-independent cytotoxic effects of PLTFBH could be explained by depletion of other HSP40/JDPs than DNAJA1. Further studies are required to clarify which other HSP40/JDPs could contribute to cytotoxic effects of PLTFBH. More importantly, it needs to be carefully determined whether PLIHZ analogs can be used for anti-cancer therapies, since they could impact the levels and activities of client proteins of other HSP40/JDPs.

Our group has recently published that DNAJA1 protein specifically binds to misfolded or conformational mutp53 but not wtp53 or DNA contact mutp53 with relatively intact p53 protein structure [20]. The biological effects of DNAJA1 in cancer cells are largely dependent on the presence of conformational mutp53; DNAJA1 promotes filopodia formation and cancer cell migration by binding to and stabilizing misfolded or conformational mutp53. These observations are in line with our finding that depletion or inhibition of DNAJA1 by PLTFBH reduces filopodia formation and migration of cancer cells specifically expressing conformational mutp53. It should also be noted that PLTFBH shows the minimal impact on the viability of non-tumor cell lines, which could suggest a therapeutic range. The pharmacological properties of PLTFBH have not yet been characterized. Following improvement of the efficacy and specificity of PLTFBH analogs to DNAJA1 and evaluation of their pharmacological properties, it is crucial to test their in vivo effects on tumor progression, as well as toxicity and safety, using pre-clinical studies. This would accelerate the development of the future DNAJA1-mutp53-targeted therapies with minimal side effects.

## 5. Conclusions

We identify PLTFBH, derived from the natural compound plumbagin, as a compound that binds to the J domain of DNAJA1, through a molecular docking study. This compound binds to and reduces protein levels of DNAJA1 as well as conformational mutp53, leading to inhibited cancer cell migration. This work highlights DNAJA1 as a therapeutic target in cancers carrying conformational mutp53, as well as the use of PLTFBH as an inhibitor of DNAJA1 and certain members of HSP40/JDPs.

## Figures and Tables

**Figure 1 cancers-14-04187-f001:**
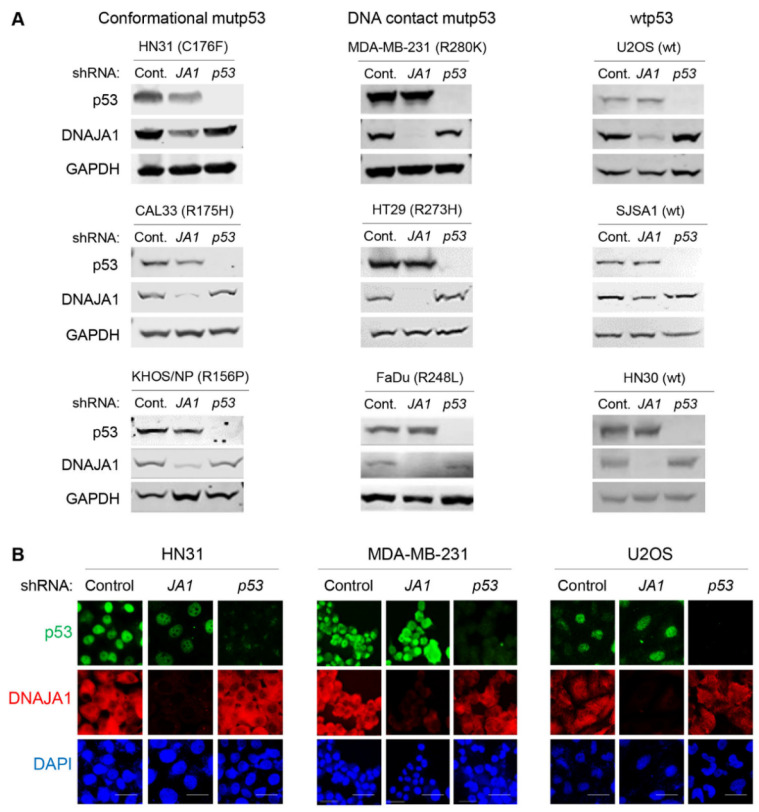
Knockdown of DNAJA1 specifically reduces protein levels of conformational mutp53, but not DNA contact mutp53 and wtp53. (**A**,**B**) Western blotting for p53, DNAJA1, and GAPDH (**A**) and immunofluorescence for p53, DNAJA1, and DAPI (**B**) using multiple cancer cells with different p53 status with or without knockdown of DNAJA1 (JA1) and p53 (p53). Scale bar: 50 µm.

**Figure 2 cancers-14-04187-f002:**
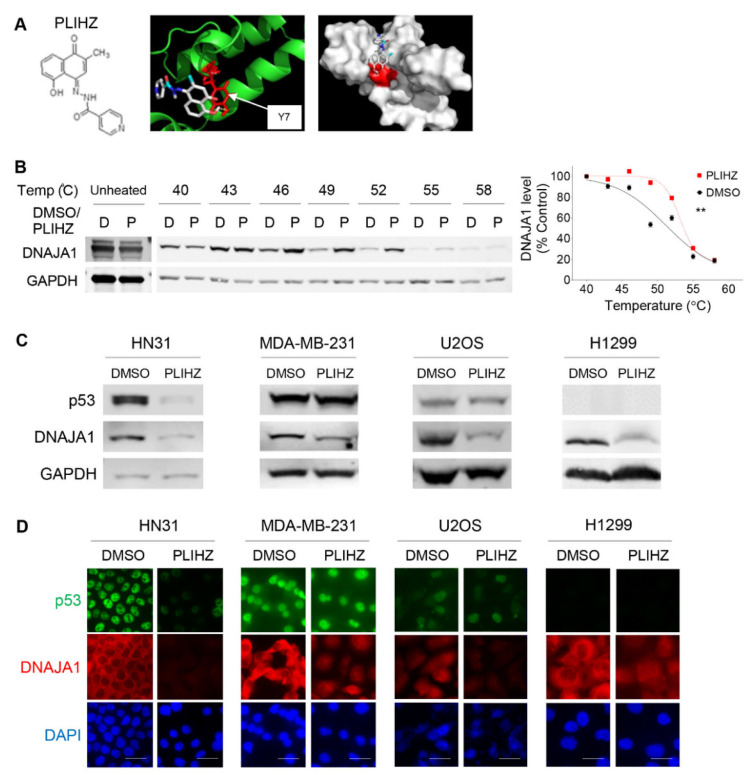
Identification of a compound that binds DNAJA1 to specifically reduce conformational mutp53. (**A**) Chemical structure of PLIHZ compound, derived from plumbagin, which was identified through a molecular docking (**left**). Ribbon and crystal structures of DNAJA1 protein (PDB: 2LOI) showing binding of DNAJA1 to PLIHZ at tyrosine residue Y7 (**right**) with a 2.6 Å bond distance and binding free energy of −6.6 kcal/mol. (**B**) CETSA showing intracellular binding of PLIHZ to DNAJA1. A representative Western blotting for DNAJA1 and GAPDH using protein extracts from CAL33 (p53^R175H^) cells with treatment with DMSO and PLIHZ at 80 µM for 4 h, followed by incubation at different temperatures for 3 min (**left**). A representative blot using protein extracts from unheated cells are also shown. A summarized graph showing normalized DNAJA1 band densities at different temperatures of 40, 43, 46, 49, 52, 55, and 58 °C (**right**). Mean ± SEM from three independent experiments (*n* = 3). ** *p* < 0.01 for two-way ANOVA. (**C**) Western blotting for p53, DNAJA1, and GAPDH using indicated cells with different p53 status, treated with DMSO or PLIHZ at ~1/2 of 24h-IC50 for 24 h. (**D**) Immunofluorescence for p53, DNAJA1, and DAPI using indicated cells treated with DMSO or PLIHZ at ~1/2 of 24h-IC50 for 24 h.

**Figure 3 cancers-14-04187-f003:**
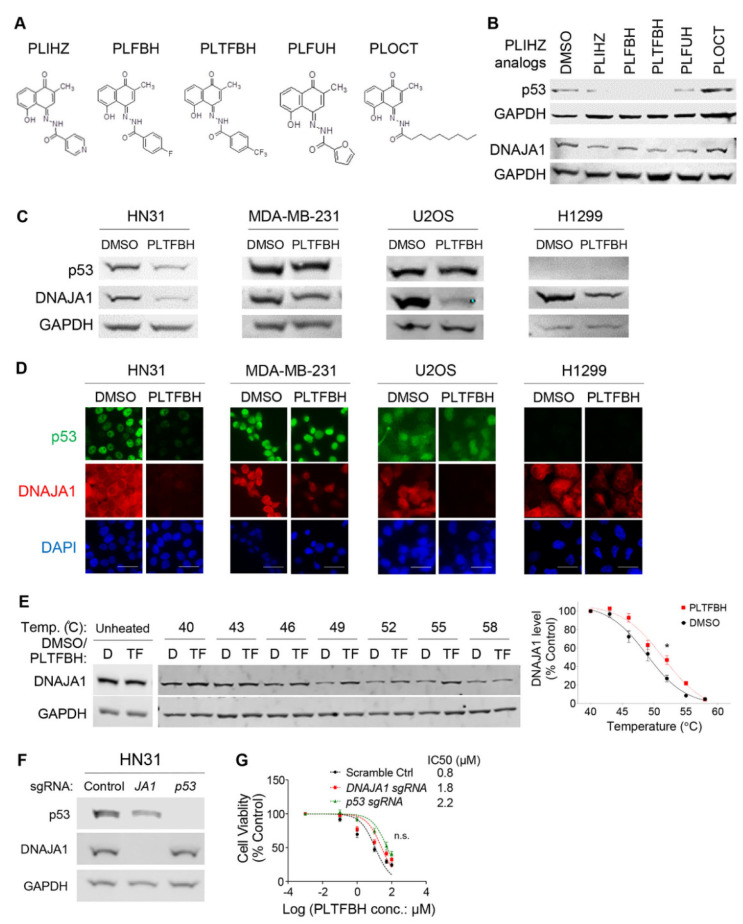
PLTFBH, an analog of PLIHZ, specifically reduces conformational mutp53 levels similar to PLIHZ. (**A**) Chemical structures of PLIHZ analogs including PLIHZ, PLFBH, PLTFBH, PLFUH, and PLOCT. (**B**) Western blotting for p53 and GAPDH using HN31 cells treated with different PLIHZ analogs (40 µM for 24 h). (**C**) Western blotting for p53, DNAJA1, and GAPDH using HN31, MDA-MB-231, and U2OS cells treated with PLTFBH at ~1/2 of 24h-IC_50_. (**D**) Immunofluorescence for p53, DNAJA1, and DAPI using HN31, MDA-MB-231, U2OS, and H1299 cells treated with PLTFBH at ~1/2 of 24h-IC_50_. Scale bar: 50 µm. (**E**) CETSA showing intracellular binding of PLTFBH to DNAJA1 in CAL33 cells: a representative Western blotting for DNAJA1 and GAPDH using protein extracts from HN31 cells (**left**); a summarized graph showing normalized DNAJA1 band densities at different temperatures (**right**). Mean ± SEM from three independent experiments (*n* = 3). * *p* < 0.05 for two-way ANOVA. (**F**) Western blotting for p53, DNAJA1, and GAPDH using HN31 cells with or without knockout for *DNAJA1* (*JA1*) or *p53* (*p53*) using the CRISPR-Cas9 strategy. (**G**) Summary of MTT assays using control, *DNAJA1* knockout, or *p53* knockout HN31 treated with different concentrations of PLTFBH for 72 h. Mean ± SEM from three independent experiments (*n* = 3). n.s.: not significant for one-way ANOVA. IC_50_ values of PLTFBH for each sub-cell line are shown on the right.

**Figure 4 cancers-14-04187-f004:**
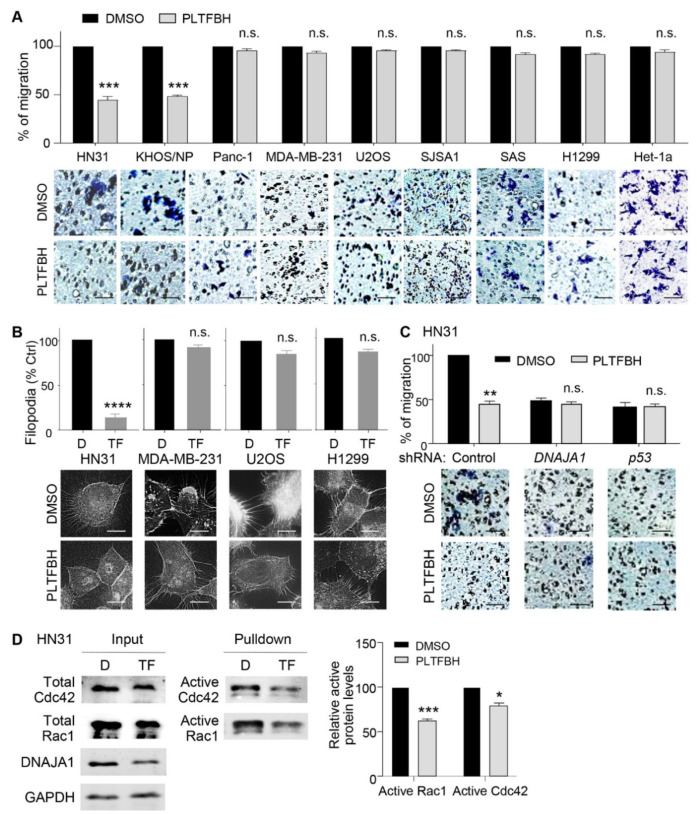
PLTFBH inhibits migratory potential of cancer cells in a manner dependent on DNAJA1 and conformational mutp53. (**A**) Transwell migration assays using indicated cells with different p53 status treated with PLTFBH at ~1/2 IC_50_ for 12 h. All cells were pre-treated with PLTFBH for 12 h, followed by trypan blue staining and transwell migration assays. **Top**: summarized graphs. **Bottom**: representative images. Mean ± SEM from three independent experiments (*n* = 3). *** *p* < 0.001 for two-tailed Student’s *t*-test. n.s.: not significant. Scale bar: 100 µm. (**B**) F-actin staining showing inhibition of filopodia formation in HN31 cells, but not MDA-MB-231, U2OS, and H1299 cells, by PLTFBH (TF). **Top:** summarized graph. **Bottom**: representative images. Mean ± SEM from three independent experiments (*n* = 3). **** *p* < 0.0001 for two-tailed Student’s *t*-test. n.s.: not significant. Scale bar: 10 µm. (**C**) Transwell migration assays using *DNAJA1-* or *p53*-knockdown HN31 cells treated with PLTFBH at ~1/2 IC_50_ for 12 h. Cells were pre-treated with PLTFBH for 12 h. Mean ± SEM from three independent experiments (*n* = 3). *** p* < 0.01 for two-tailed Student’s *t*-test. n.s.: not significant. Scale bar: 100 µm. (**D**) Rac1/Cdc42 activation assays following pulldown of active Rac1 and Cdc42 using protein extracts from HN31 cells treated with DMSO (D) or PLTFBH (TF) at ~1/2 of 24h-IC_50_. **Left**: representative immunoblots. **Right**: summarized graph. Mean ± SEM from three independent experiments (*n* = 3). * *p* < 0.05, *** *p* < 0.001 for two-tailed Student’s *t*-test.

**Figure 5 cancers-14-04187-f005:**
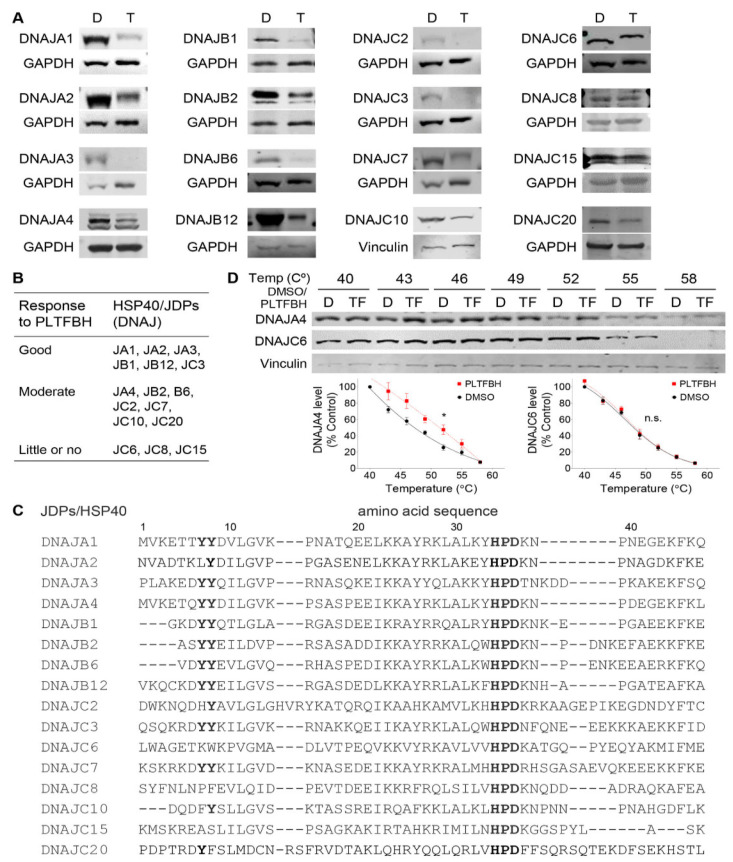
PLTFBH selectively decreases protein levels of certain members of HSP40/JDPs. (**A**) Western blotting for several members of HSP40/JDPs and GAPDH using HN31 cells treated with DMSO (D) or PLFBH (TF) at ~1/2 IC50 for 24 h. (**B**) Three groups (good, moderate, little or no) of HSP40/JDPs based on their response to PLTFBH. Note that some blots were from the same membrane (DNAJB1 and DNAJB2; DNAJB6 and DNAJC6; DNAJC2 and DNAJC3), hence using the same GAPDH control. (**C**) Amino acid sequence alignment of the J domain of multiple HSP40/JDPs by centering the HPD sequence. (**D**) CETSA showing intracellular binding of PLTFBH with DNAJA4, but not DNAJC6, in CAL33 cells treated with 80 µM of PLTFBH (TF) for 4 h. Vinculin was used as a loading control. Summarized graphs showing normalized DNAJA4 and DNAJC6 band densities at different temperatures (**right**). Mean ± SEM from three independent experiments (*n* = 3). * *p* < 0.05 for two-way ANOVA.

**Figure 6 cancers-14-04187-f006:**
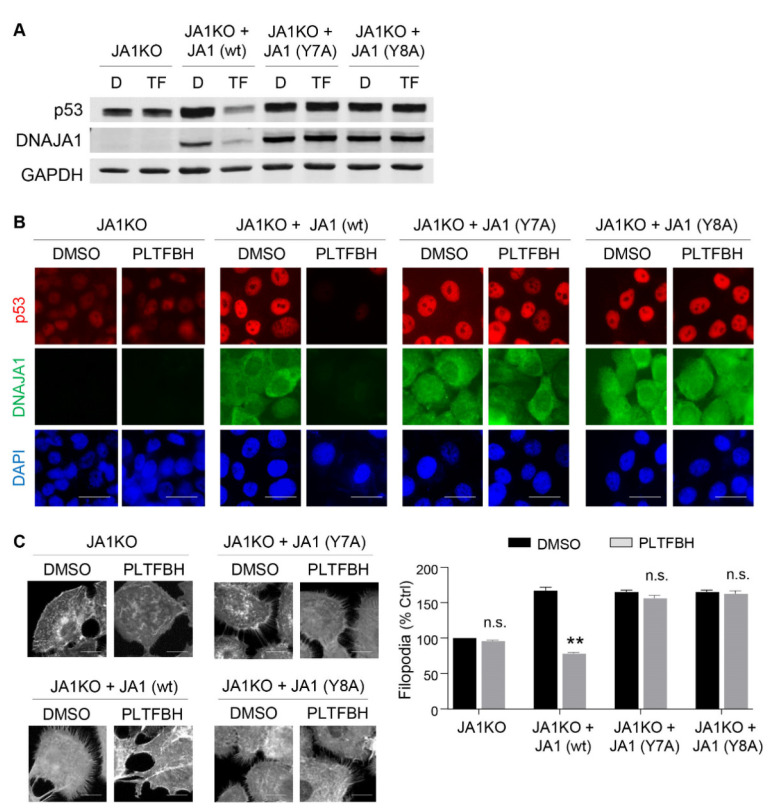
Mutations at Y7 and Y8 residues in DNAJA1 abrogate the ability of PLTFBH to deplete DNAJA1 and conformational mutp53. (**A**) Western blotting to detect exogenous DNAJA1 and GAPDH, as well as endogenous p53 (p53^C176F^), using *DNAJA1*-knockout HN31 cells (JA1KO) expressing exogenous wild-type (wt), Y7A mutant (Y7A), and Y8A mutant (Y8A) DNAJA1, treated with DMSO (D) or PLTFBH (TF) at ~1/2 IC_50_ for 24 h. (**B**) Immunofluorescence for p53, DNAJA1, and DAPI using the same experimental set of HN31 sub-cell lines as in Figure 6A. Scale bar: 50 µm. (**C**) F-actin staining using the HN31 sub-cell lines treated with DMSO or PLTFBH at ~1/2 IC_50_ for 24 h. Mean ± SEM from three independent experiments (*n* = 3). ** *p* < 0.01 for two-tailed Student’s *t*-test. n.s.: not significant. Scale bar: 10 µm. **Left**: representative images. **Right**: summarized graph.

## Data Availability

The data presented in this study are available on request from the corresponding author.

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
