# Peer review of "Mutant p53 Depletion by Novel Inhibitors for HSP40/J-Domain Proteins Derived from the Natural Compound Plumbagin"

_cancers, 2022, doi:10.3390/cancers14174187_

Round 1
Reviewer 1 Report
Dominant-negative and gain-of-function effects of mutant p53 are dependent on elevated protein levels, therefore drugs that induce degradation of abnormal p53 protein, such as HSP90 inhibitors or anti-aggregation peptides are considered potential anti-cancer therapeutics. In the manuscript “Mutant p53 depletion by novel inhibitors for HSP40/J-domain 2 proteins derived from the natural compound plumbagin” Alalem and colleagues report identification and characterization of plumbagin derivatives PLIHZ and PLTFBH as compounds that can destabilize mutant p53 via inhibition of DNAJA1, a member of HSP40 protein family. Using in silico modeling authors identified plumbagin from a natural compound-derived small molecule library. Further, they generated more potent plumbagin derivatives and tested the effects of these compounds on conformational mutant p53 protein levels in cancer cells. The authors showed that the compounds specifically inhibited migration of cancer cells with structural mutant p53.
This work identifies a novel inhibitor for HSP40/JDP, which is interesting and relevant. The manuscript is well written, the experimental approaches are adequate and provide sufficient data supporting the finding. The study can be published without major revision. There are some minor issues:
1. Figure 2B, right panel. The graph is too small to recognize the labels properly. It is recommended to use colors to distinguish two curves.
2. Figure 2D. The micrograph showing IF staining in HN31 cells demonstrates nuclear localization of DNAJA1, which is inconsistent with other figures (1B, 3D) showing cytoplasmatic distribution of DNAJA1 in the same type of cells.
3. Figure 4A is missing in the text (section 3.4).
Author Response
Thank you very much for reviewing our manuscript. The following is our response.
1. Figure 2B, right panel. The graph is too small to recognize the labels properly. It is recommended to use colors to distinguish two curves.
Accordingly, we have used colors and also made the graph enlarged.
2. Figure 2D. The micrograph showing IF staining in HN31 cells demonstrates nuclear localization of DNAJA1, which is inconsistent with other figures (1B, 3D) showing cytoplasmatic distribution of DNAJA1 in the same type of cells.
We appreciate the reviewer’s observation. We have now shown better representative images in HN31 immunofluorescence in Figure 2D, showing that DNAJA1 is mainly localized to the cytoplasm.
3. Figure 4A is missing in the text (section 3.4).
Our apology for the mistake. We have corrected it.
Reviewer 2 Report
In this manuscript from Dr. Tomoo Iwakuma’s group, authors identified PLTFBH, derived from the natural compound plumbagin, as a compound that binds to the J domain of DNAJA1, through a molecular docking study. Their study showed that this compound binds to and reduces protein levels of DNAJA1 as well as conformational mutant p53 (mutp53), leading to inhibited cancer cell migration. This work highlights DNAJA1 as a therapeutic target in cancers carrying conformational mutp53. The finding of this study is novel and significant given that p53 is frequently mutated in human cancer and accumulation mutp53 in tumors is crucial for malignant progression. The experiments are well designed and performed. Overall results are strong and support the main conclusion of this manuscript.
Minor comments:
1. In this manuscript, statistical analysis only used p < 0.05. Authors are suggested to use different p values to present the statistical significance. Number (n) for analysis is expected to be presented. In addition, in Fig 6, no p value was shown for statistical analysis.
2. It will be more informative and helpful for the mutp53 research field if authors can discuss the potential application of PLTFBH for future studies using animal models.
Author Response
Thank you very much for reviewing our manuscript. Please see below response.
1. In this manuscript, statistical analysis only used p < 0.05. Authors are suggested to use different p values to present the statistical significance. Number (n) for analysis is expected to be presented. In addition, in Fig 6, no p value was shown for statistical analysis.
Accordingly, we have also added **, p<0.01; ***, p<0.001; ****, p<0.0001 and indicate the “n” number in each panel when necessary (mostly three-independent experiments (n=3)), in addition to describing the statistical analysis in the legend of every experiment.
2. It will be more informative and helpful for the mutp53 research field if authors can discuss the potential application of PLTFBH for future studies using animal models.
We totally agree with the reviewer’s comment. The pharmacological properties of PLTFBH have not yet been characterized. Following improvement of the efficacy and specificity of PLTFBH analogs to DNAJA1 and evaluation of their pharmacological properties, it is crucial to test their in vivo effects on tumor progression, as well as toxicity and safety, using pre-clinical studies. We have mentioned it in the discussion.